# Evaluation of the Analytical Performance of a Lateral Flow Assay for the Detection of Anti-*Coccidioides* Antibodies in Human Sera—Argentina

**DOI:** 10.3390/jof10050322

**Published:** 2024-04-28

**Authors:** Mariana N. Viale, Diego H. Caceres, Patricia E. Mansilla, María C. Lopez-Joffre, Flavia G. Vivot, Andrea N. Motter, Adriana I. Toranzo, Cristina E. Canteros

**Affiliations:** 1Laboratorio Nacional de Referencia en Micología Clínica, Departamento Micología, Instituto Nacional de Enfermedades Infecciosas (INEI), Administración Nacional de Laboratorios e Institutos de Salud (ANLIS) “Dr. Carlos G. Malbrán”, Buenos Aires 1282, Argentina; mcecilj@hotmail.com (M.C.L.-J.); fgvivot@anlis.gob.ar (F.G.V.); atoranzo@hotmail.com (A.I.T.); cecanteros@gmail.com (C.E.C.); 2Center of Expertise in Mycology Radboudumc/CWZ, 6525GA Nijmegen, The Netherlands; diegocaceres84@gmail.com; 3Studies in Translational Microbiology and Emerging Diseases (MICROS) Research Group, School of Medicine and Health Sciences, Universidad del Rosario, Bogota 111221, Colombia; 4Immuno-Mycologics Inc. (IMMY), Norman, OK 73069, USA; 5Hospital Interzonal San Juan Bautista, San Fernando del Valle de Catamarca 4700, Argentina; p.evangelina.mansilla@gmail.com; 6Unidad Operativa Centro de Contención Biológica, Administración Nacional de Laboratorios e Institutos de Salud (ANLIS) “Dr. Carlos G. Malbrán”, Buenos Aires 1282, Argentina; andreamotter@gmail.com

**Keywords:** coccidioidomycosis, *Coccidioides*, human, antibody, immunodiagnostic, lateral flow assay, enzyme immunoassay, immunodiffusion, counterimmunoelectrophoresis, Argentina

## Abstract

Coccidiomycosis is a potentially life-threatening fungal infection endemic to certain regions of Argentina. The infection is caused by *Coccidioides* spp. and is primarily diagnosed by *Coccidioides* antibody (Ab) detection. Access to rapid, highly accurate diagnostic testing is critical to ensure prompt antifungal therapy. The sōna *Coccidioides* Ab Lateral Flow Assay (LFA) performs faster and requires less laboratory infrastructure and equipment compared with other Ab detection assays, potentially providing a substantial improvement for rapid case screening in coccidioidomycosis-endemic regions; however, validation of this test is needed. Thus, we aimed to evaluate the analytical performance of the sōna *Coccidioides* Ab (LFA) and compare agreement with anti-*Coccidioides* Ab detection assays. A total of 103 human sera specimens were tested, including 25 specimens from patients with coccidioidomycosis and 78 from patients without coccidioidomycosis. The sōna *Coccidioides* Ab Lateral Flow Assay (LFA) was performed with a sensitivity of 88%, and specificity and accuracy of 87%. Furthermore, the *Coccidioides* Ab LFA had good agreement with other anti-*Coccidioides* Ab detection assays. Our findings suggest the sōna *Coccidioides* Ab LFA has satisfactory performance and may be useful for diagnosing coccidioidomycosis in endemic regions.

## 1. Introduction

Coccidioidomycosis is a fungal disease caused by *Coccidioides immitis* and *C. posadasii*. Infection is the result of exposure to contaminated soil, primarily by inhalation of fungal arthroconidia found in the matter. The mycosis predominantly affects individuals who reside in endemic regions, which are localized in the southwestern United States, Northern Mexico, Guatemala, Honduras, Venezuela, Northeastern Brazil, Southern Bolivia, Western Paraguay, and the arid pre-Andes region of Argentina [1,2,3,4].

Endemic regions in Argentina are characterized by arid to semi-arid climates, in the arid foothills from the province of Jujuy to the province of Río Negro, with the highest number of cases reported in the province of Catamarca [5]. Occupational exposure, particularly during agricultural activities and construction, is associated with a higher risk of infection [3,6,7,8,9,10]. Other activities such as digging, excavation, and disturbance of soil can lead to the inhalation of arthroconidia. Furthermore, climatic variations and environmental changes may influence the distribution of *Coccidioides* [11,12,13,14]. Surveillance efforts in Argentina, like the mandatory reporting of cases in endemic regions like Catamarca and the recent incorporation of national-level reporting, aim to monitor epidemiological trends of the disease, identify high-risk areas, and enhance public awareness [15,16].

Reference diagnosis of coccidioidomycosis is performed by microscopic observation of the fungus and culture. Handling of culture specimens is complex due to the high biohazard risk; microscopic analysis is rapid, but the performance of this test is variable. Antibody (Ab) detection using complement fixation (CF), immunodiffusion (ID), counterimmunoelectrophoresis (CIE), and enzyme immunoassay (EIA) is widely adopted, particularly due to the use of minimally invasive specimens and their commendable analytical performance. However, a notable limitation of these assays is the extended turnaround time required to obtain antibody detection results [1,17]. For these reasons, accurate assays are needed. The principal goal of this study was to evaluate the analytical performance of a novel lateral flow assay (LFA) for the rapid detection of anti-*Coccidioides* antibodies in human sera.

## 2. Materials and Methods

Specimens: 103 human serum specimens were tested. Coccidioidomycosis specimens include 25 sera, these sera came from 12 patients with proven coccidioidomycosis by culture (*n* = 17 sera), and 8 patients with positive antibody testing for coccidioidomycosis (*n* = 8 sera). We also tested 78 sera from persons without coccidioidomycosis, including 30 sera specimens from people with other etiologies for respiratory disease who live in endemic regions for coccidioidomycosis. Also, we included 18 sera from patients with positive antibody testing for histoplasmosis, 16 from patients with positive antibody testing for paracoccidioidomycosis, and 14 from patients with positive antibody testing for aspergillosis (Figure 1).

Specimens were remnants from serological studies conducted at the National Reference Laboratory in Clinical Mycology of Argentina (LNRM). These specimens were stored at −20 °C, in single-use aliquots, and were collected between 2009 and 2021. We tested all available sera from patients with proven and positive antibody testing for coccidioidomycosis, and sera from patients with positive antibody testing against other fungi, *Histoplasma*, *Paracoccidioides*, and *Aspergillus*, as specificity controls. Prior to EIA and LFA testing, specimens were re-tested by ID and CIE methods to validate specimen stability and the presence of anti-*Coccidioides* antibodies. (Figure 1).

*Coccidioides* Antibody (Ab)-detection Assays: four assays were evaluated. Two in-house assays, including CIE and ID. These assays were performed using an in-house *Coccidioides* antigen (Ag), and a rabbit serum containing polyclonal Ab for positive control, both produced by the LNRM [18]. Two commercial immunodiagnostic assays from IMMY (Norman, OK, USA) were also used: the Clarus *Coccidioides* Ab Enzyme Immunoassay (*Coccidioides* Ab EIA) and the sōna *Coccidioides* Ab Lateral Flow Assay (*Coccidioides* Ab LFA).

*Coccidioides* Ab CIE was performed following laboratory standard operating procedure (SOP). First, gel plates for CIE were prepared using 1% agarose tipe II medium EEO (Sigma, St Louis, USA)/Tris Acetate-EDTA (TAE) buffer. Wells were created using a puncher and were filled with the specimens (8 µL), the Ag (2.5 µL), and the positive control (8 µL). Electrophoresis was performed in a TAE buffer at 100 V for 90 min. Results were interpreted by the visualization of precipitin bands in the plates.

*Coccidioides* Ab ID was performed following laboratory SOP. In brief, 20 µL of human sera, *Coccidioides* Ag, and positive control sera were placed into 1% phenolyzed agar ID plates. Plates were incubated at room temperature, and results were interpreted after 72 h. A positive result was interpreted as the presence of precipitin identity bands in the ID plates. Standard operating procedure (SOP) available at http://www.anlis.gov.ar/inei/micologia/?page_id=1000 (accessed on 18 March 2024).

*Coccidioides* Ab EIA was achieved following the instructions described in the kit package insert. Briefly, 100 µL of diluted 1:441 sera, CF and TP calibrator cutoff, and kit positive and negative controls were placed into microwells coated with CF or TP *Coccidioides* antigen, respectively. Plates were incubated at 20–25 °C (room temperature) for 30 min. After incubation, plates were manually washed three times using 1× wash buffer. Then, 100 µL of CF or TP enzyme conjugate (HRP) was added to their respective plates. Plates were incubated for 30 min at room temperature. Following the incubation, plates were washed again three times as previously described. Then, 100 µL of TMB Substrate was added to each well and incubated in darkness for 10 min at 20–25 °C. Finally, 100 µL of stop solution was dispensed to all wells. Plate’s optical densities (OD) were obtained using a dual-wavelength plate reader at 450 and 630 nm using the Labsystems Multiskan RC spectrophotometer (Thermo Electron Corporation, Vantaa, Finland). EIA units were obtained by dividing the OD value of serum and control wells by the blanked OD from the calibrator cutoff well. Specimens with EIA unit values ≥1.5 were considered positive, specimens with EIA unit values between ≥1.0 to 1.49 were considered indeterminate, and specimens with EIA unit values <1.0 were considered negative. For the 2 × 2 table analysis, we considered indeterminate as negatives.

*Coccidioides* Ab LFA was carried out following manufacturer instructions. Sera specimens were diluted 1:441 using kit specimen diluent. LFA strip was placed in the tube containing 100 µL of the diluted specimen, and results were interpreted after 30 min of incubation at room temperature. The presence of two pink or red lines, at the control and test zone, was considered a positive result. The presence of only the control line was considered a negative result. The absence of the control line invalidated the result, regarding the presence or the absence of the test line.

Statistical analysis: Test performance and respective 95% confidence intervals (95% CI) were calculated using 2 × 2 tables. Concordance analysis was conducted by calculation of Kappa. Kappa values were interpreted as 0.0 to 0.2, no agreement; 0.21 to 0.39, minimal agreement; 0.40 to 0.59, weak agreement; 0.60 to 0.79, moderate agreement; 0.80 to 0.89, strong agreement; and ≥0.90, almost perfect agreement. Data were analyzed using STATA 11.

## 3. Results

Characteristics of patients with coccidioidomycosis: A total of 20 patients with coccidioidomycosis were included, comprising 11 females and 9 males, with ages ranging from 10 to 73 years old. The participants were primarily from Catamarca (*n* = 12). Regarding baseline conditions, nine patients had no reported conditions, three had HIV, two had diabetes, two had cancer, and four had no available information. Twelve patients presented pulmonary coccidioidomycosis, subacute and chronic forms, five had disseminated coccidioidomycosis, and three had no specified information (Table 1).

*Coccidioides* Ab LFA: Of the 25 sera from patients with coccidioidomycosis, 22 were positive for *Coccidioides* Ab LFA, showing a sensitivity of 88% (95% CI: 69–97). Of the three false negative specimens, one was from a patient with advanced HIV disease; this patient had proven coccidioidomycosis by culture (*C. posadasii*) and also had positive IgM EIA (3.1 EIA units). The second specimen was from a probable case, with positive CIE and IgM EIA (2.4 EIA units). The third specimen was from a culture-proven case (*C. posadasii*), with positive CIE, ID, and IgG EIA (4.1 EIA units). In the 78 controls without coccidioidomycosis, 10 specimens tested positive by the LFA, showing a specificity of 87% (95% CI: 78–94). Most of the false positive results were found in sera from people with paracoccidioidomycosis, *n* = 7 (7 out of 16 sera, 44% cross-reactivity), one false positive in a patient with aspergillosis (1 out of 14 sera, 7% cross-reactivity), one false positive in a patient with histoplasmosis (1 out of 18 sera, 6% cross-reactivity), and one false positive in sera from a person with respiratory disease who live in a endemic region for coccidioidomycosis (1 out of 30, 3% false positivity). Assay accuracy was 87% (95% CI: 79–93) (Table 2A, and Figure 2).

We also evaluate the performance of the other assays for the anti-*Coccidioides* Ab detection. The performance of these assays is described below.

*Coccidioides* Ab CIE: In total, 23 of the 25 sera from individuals with coccidioidomycosis had positive results for CIE (sensitivity 92%, 95% CI: 74–99). The two false negatives were proven cases. One serum was a proven case that tested positive by the LFA only. The second specimen was a proven case with positive IgM EIA only (HIV patient described above). On the other hand, 77 out of the 78 non-coccidioidomycosis control specimens were negative (99% specificity, 95% CI: 93–100). The only false positive result was found in a patient with proven paracoccidioidomycosis. The CIE shows an accuracy of 97% (95% CI: 92–99) (Table 2B, and Figure 2).

*Coccidioides* Ag ID: In total, 17 out of 25 sera tested positive (sensitivity 68%, 95% CI: 47–85). False negatives involved six specimens from proven cases, and two specimens from probable cases. Four sera from proven cases that tested positive for CIE, LFA, and IgG EIA. One serum from a probable case that tested positive by CIE, LFA, IgM, and IgG EIA. One specimen from a proven case tested positive by the IgM EIA only (patient with HIV described before). And one serum from a proven case that tested positive only by the LFA. No false positive results were recorded using ID (specificity of 100%; 95% CI: 95–100). ID displayed an assay accuracy was 92% (95% CI: 85–97) (Table 2C, and Figure 2).

*Coccidioides* Ab EIA: IgM EIA was performed with 36% sensitivity, 9 positive out of 25 coccidioidomycosis cases. IgG was performed with a sensitivity of 88%, with 22 positives out of 25 cases (the 3 discrepant results were indeterminate). Combining IgM and IgG, performance was 96%, showing one false negative result. This IgM/IgG false negative was serum from a proven case, with indeterminate IgG (1.0 EIA units), negative IgM, positive LFA, and negative CIE and ID. In terms of assay specificity, two false positives were observed using IgM EIA (IgM specificity of 97%; 95% CI: 91–100), these false positives were found in sera from patients with histoplasmosis (cross-reaction). No false positives were observed using IgG EIA. Combining IgM/IgG EIA, we found a specificity of 97% (95% CI: 91–100). This specificity was impacted by the histoplasmosis cross-reactions observed using IgM (Table 2D, and Figure 2). It is important to highlight that in control specimens the presence of indeterminate results was frequent. We observed 7 sera with IgM indeterminate result and 12 sera with IgG indeterminate result. However, the ROC analysis did not show an alternative cut-off that improved EIA performance, the reason why we kept the cut-off recommended by the manufacturer.

Concordance analysis: The concordance between all assays ranged from moderate to very good. The majority of assays were performed with good agreement (Figure 3).

## 4. Discussion

Most of the coccidioidomycosis cases are diagnosed using Ab detection assays, based on complement fixation (CF), gel precipitation tests (CIE and ID), and EIA. The main limitation of these assays is the need for highly trained laboratorians, special laboratory instruments, and prolonged turnaround time for results [19]. LFA technology has significantly improved this limitation; these assays are simple, accurate, and rapid. In this study, we were able to evaluate and compare a LFA for the detection of anti-*Coccidioides* Ab in human sera, sōna *Coccidioides* Ab LFA (IMMY^TM^, Norman, OK, USA). Results here showed high analytical performance and good correlation with other assays used for the immunodiagnosis of coccidioidomycosis.

The sōna *Coccidioides* Ab LFA was performed with a sensitivity of 88%. In terms of sensitivity, these results are comparable to results reported by Contreas et al. at the University of California, in Los Angeles, CA, USA (poster ASM meeting 2019), but are substantially different from results reported by Donovan et al. at the University of Arizona, AZ, USA [20,21]. In the present study, it is important to highlight that in two of three false negative LFA specimens, we observed positive IgM results and negative IgG results, suggesting early disease in false negatives. Another factor could be the high levels of immunosuppression, as observed in patients with advanced HIV disease. These factors might have affected the concentration of anti-*Coccidioides* Ab, which was likely below the limit of detection of the LFA [22]. In addition, the main substantial difference of this study compared with the report of Donovan et al. is the low frequency of patients with acute coccidioidomycosis [21]. Specimens tested here were mainly from patients with chronic and disseminated coccidioidomycosis. That could explain the higher LFA sensitivity in this study.

We also evaluated the performance of the sensitivity of other immunodiagnostic assays. Based on that analysis, we found that the combination of IgM and IgG detection was the assay with the highest sensitivity, followed by the CIE. ID was 68% sensitivity, and the detection of IgM alone was the assay with the lowest sensitivity (36%).

Regarding specificity, we observed that cross-reactivity with sera from patients with paracoccidioidomycosis was the most frequent discrepancy. We also observed a lower frequency of false positive results in sera from patients with aspergillosis and histoplasmosis. It is well known that cross-reactions, especially in specimens from individuals with fungal infections, could happen. One false positive result was observed in serum from a patient with non-coccidioidomycosis respiratory disease, this patient resided in a endemic region for coccidioidomycosis, probably the positive results was a consequence of the environmental exposure to *Coccidioides* product and of living in an endemic region. By eye reading, differences in band intensity are difficult to interpret, a reason why the addition of a LFA reader or other alternatives for automatic reading of LFA devices could improve the interpretation and performance of the assay.

Simplicity, rapid testing, and high accuracy are characteristics that make the sōna *Coccidioides* Ab Lateral Flow Assay a desirable alternative for rapid screening of possible coccidioidomycosis cases. But, compared with the other Ab assays evaluated in this study, the *Coccidioides* Ab LFA performed with the lowest specificity. For this reason, it would be important to confirm positive LFA results by the use of a confirmatory assay, such as the *Coccidioides* Ab EIA, or the *Coccidioides* Ab CIE, given the high diagnostic analytical performance of these assays. This finding was previously reported, and similar recommendations have been proposed [23].

Principal limitations of this study were the small sample size, the use of sera from some probable cases, the limited access to charts of cases, the unblinded testing of specimens, the lack of an automatic reader for LFA interpretation, and the lack of sera specimens of asymptomatic people who are residents and not residents in the coccidioidomycosis-endemic region. Another major limitation is the lack of specimens from patients with acute pulmonary coccidioidomycosis.

## 5. Conclusions

Here, we evaluate a novel LFA for the detection of anti-*Coccidioides* antibodies and were able to evaluate the analytical performance of other assays for the immunodiagnosis of coccidioidomycosis. Diagnosis of coccidioidomycosis could take days to hours using CF, gel precipitation tests, and EIA; the use of LFAs reduces this time to minutes. Another advantage of LFAs is the simplicity of testing implementation in low-complexity laboratories located in regions where the disease is endemic, with limited access to reference laboratories.

## Figures and Tables

**Figure 1 jof-10-00322-f001:**
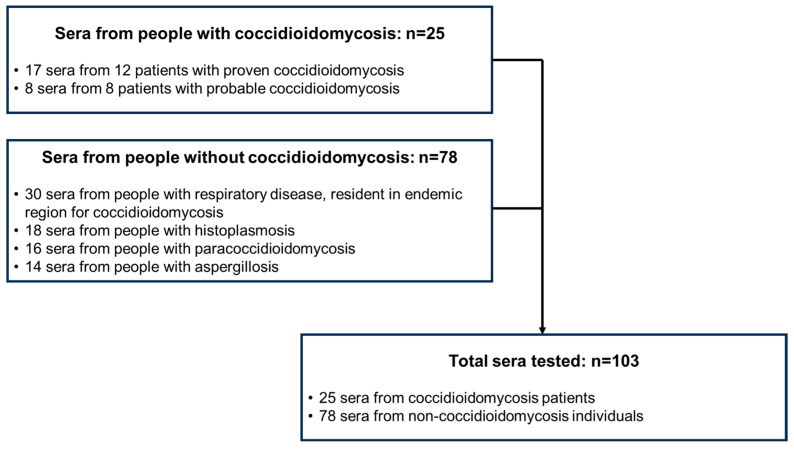
Flow chart of specimens tested for the evaluation of the analytical performance of the sōna *Coccidioides* Ab Lateral Flow Assay in human sera.

**Figure 2 jof-10-00322-f002:**
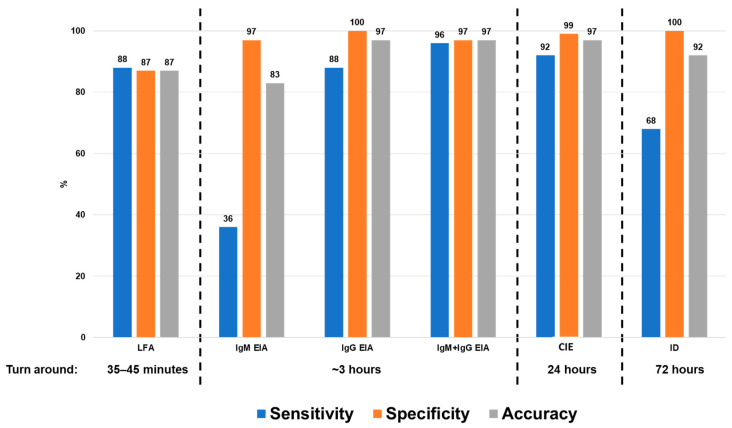
Comparison of turn-around of results and sensitivity, specificity, and accuracy of assay for the detection of *Coccidioides* Ab.

**Figure 3 jof-10-00322-f003:**
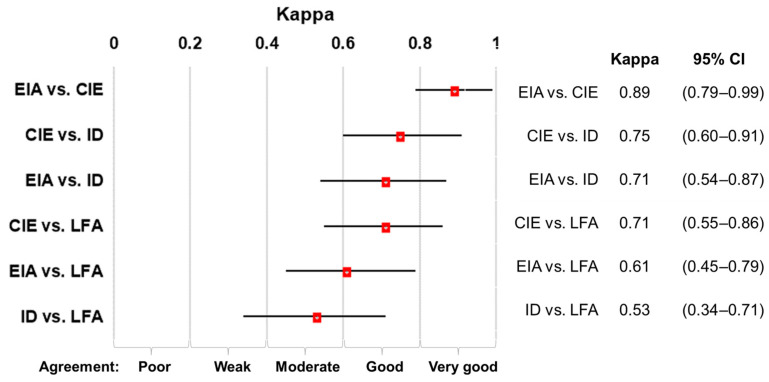
Concordance analysis of assays for the detection of *Coccidioides* Ab. The red boxes correspond to the kappa value, the black lines represent the kappa confidence intervals (CI).

**Table 1 jof-10-00322-t001:** Characteristics of patients with coccidioidomycosis.

Male	*n* = 9
Female	*n* = 11
Age range	10 to 73 years old
**Residence**	
Catamarca	*n* = 12
Salta	*n* = 3
Santiago del Estero	*n* = 2
Mendoza	*n* = 1
La Rioja	*n* = 1
Jujuy	*n* = 1
**Baseline conditions**	
No reported	*n* = 9
HIV	*n* = 3
Diabetes	*n* = 2
Cancer	*n* = 2
No information	*n* = 4
**Coccidioidomycosis clinical form**	
Pulmonary (subacute or chronic)	12 *
Disseminated	5
No information	3

(*) no acute pulmonary coccidioidomycosis cases recorded.

**Table 2 jof-10-00322-t002:** Analytical performance of assays for detection of *Coccidioides* Ab in human sera.

**(A) *Coccidioides* Ab LFA**			
		Coccidioidomycosis	
		+	−
LFA	+	22	10
	−	3	68
Statistic		Value (%)	95% CI
Sensitivity		88	69–97
Specificity		87	78–94
Accuracy		87	79–93
**(B) *Coccidioides* Ab CIE**			
		Coccidioidomycosis	
		+	−
CIE	+	23	1
	−	2	77
Statistic		Value (%)	95% CI
Sensitivity		92	74–99
Specificity		99	93–100
Accuracy		97	92–99
**(C) *Coccidioides* Ab ID**			
		Coccidioidomycosis	
		+	−
ID	+	17	0
	−	8	78
Statistic		Value (%)	95% CI
Sensitivity		68	47–85
Specificity		100	95–100
Accuracy		92	85–97
**(D) *Coccidioides* Ab EIA (IgM+IgG)**			
		Coccidioidomycosis	
		+	−
EIA	+	24	2
	−	1	76
Statistic		Value (%)	95% CI
Sensitivity		96	80–100
Specificity		97	91–100
Accuracy		97	92–99

(LFA) lateral flow assay; (CIE) counterimmunoelectrophoresis; (ID) immunodiffusion; (EIA) enzyme immunoassay assay.

## Data Availability

Data are contained within the article.

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
