# Peer review of "Evaluation of the Analytical Performance of a Lateral Flow Assay for the Detection of Anti-*Coccidioides* Antibodies in Human Sera—Argentina"

_jof, 2024, doi:10.3390/jof10050322_

Round 1

Author Response

Reviewer 1

Major comments

Summary: This study describes the diagnostic performance of multiple assays used to detect anti-Coccidioides antibodies in a cohort of stored serum samples that are incompletely clinically characterized. Additional information and major revisions are required before publication can be recommended. There are grammatical errors that will need corrected during editorial review.

R. Thank you for your comments, based on, we did modifications in the manuscript.

Detail comments

General Comments:  Readers need a better description of how samples were selected.  Line 134 says they were acquired from a biobank. How large was the cohort of samples these were selected from?  What were the methods/criteria used to select these samples? How many samples in the biobank were not used and why? This is a potential source of selection bias.  Following STARD guidelines, or something similar is recommended. A proper STARD diagram would also be appropriate – at least for the LFA.

R. Following your recommendation we added the following statement: Specimens were remnants from serological studies conducted at the National Reference Laboratory in Clinical Mycology of Argentina. These specimens were stored at −20 °C, in single-use aliquots, and were collected between 2009 and 2021. We tested all available sera from patients with probable/proven coccidioidomycosis and some sera from patients with antibodies against other fungi as specificity controls. Prior to EIA and LFA testing, specimens were re-tested by ID and CIE methods to validate specimen stability, and the presence of anti-Coccidioides antibodies.

Additional information for the cases would be useful. Were these pulmonary (acute vs. chronic), disseminated, or other cases. Were they immunosuppressed? This is mentioned for a few cases, but not for many. This will help the reader/physician assess external validity related to their patient population.

R. We added a new table describing the characteristics of the patients with coccidioidomycosis evaluated.

Title: This is not a “validation” study. Estimated diagnostic performance is one aspect of validation but this study lacks most of the analytical components. I would remove the claim that this study “validated” the LFA throughout the manuscript.

R. Based on your comment we modified the manuscript title and study goal.

Line 63: Only one sample per patient should be used for the description of diagnostic performance. This fits the implied clinical use for diagnosis. If the purpose of this study was to investigate the assays’ use overtime (treatment monitoring, screening over time, etc.) then including multiple samples from the same patients might be indicated.

R. Due to the limited number of specimens available we decided to evaluate all specimens available.

Line 66: What endemic region(s) were the patients from?

R. Based on this and comments above we include a new table describing the characteristics of coccidioidomycosis cases.

Line 68: Need additional storage and handling information. How long were sera stored at -80C? How many freeze-thaw cycles? Is there sample stability data to support including these samples?

R. Specimens were stored at -20C, specimens were aliquot for one time testing. Prior to EIA and LFA testing, specimens were re-tested by ID and CIE methods to validate specimen stability. We added this description in the material and methods section.

Line 80: Consider ‘diagnoses’ for ‘diagnosis’ OR ‘was’ for ‘were’

R. Thank you. We changed it based on your recommendation.

Line 80: How was the diagnosis of coccidioidomycosis ruled out?

R. We used case definition following EORCT/MSG and CDC guidelines.

Line 82: Since the EORTC guidelines were used, how many were proven vs. probable cases for the non-cocci groups? What were the respiratory diagnosis for the non-fungal group.

R. Yes, other fungal infections were probable. We used specimens with positive Ab testing for other mycoses aimed to confirm assay specificity. On the other hand, information from people with other etiologies for respiratory disease who live in endemic region for coccidioidomycosis was limited but they were being studied for community-acquired pneumonia.

Line 83: Were investigators performing the assays blinded to the results of all past and current (this study) diagnostic test results and group classifications?

R. No, assays were not done blindly.

Line 90: The fact that a laboratory SOP was used is not informative to the reader. Is the test or components and SOP (instructions for use) approved by a regulatory body? Is the test commercially available? If the SOP is important for the reader, consider including it as supplemental information.

R. We added the website of the National Reference Laboratory in Clinical Mycology of Argentina with the SOP recommend to the Argentinean National Mycology Laboratory Network.

Line 93: How was positive control used for interpretation?

R. We used positive control for classification of bands. We only report, and find, bands with identity.

Line 96: The fact that a laboratory SOP was used is not informative to the reader. Is the test or components and SOP (instructions for use) approved by a regulatory body? Is the test commercially available? If the SOP is important to the reader, consider including it as supplemental information.

R. We added the website of the National Reference Laboratory in Clinical Mycology of Argentina with the SOP recommend to the Argentinean National Mycology Laboratory Network.

Line 97: Which Coccidioides antigen(s) were used? Was it CF and TP antigens?

R. CF antigen only.

Line 99: How was the control serum used for interpretation?

R. We used positive control for classification of bands. We only report, and find, bands with identity.

Line 101: See comments below (Line 116)

Line 101: Specific plate washer if used and spectrophotometer (manufacturers) should be reported. The reader should be able to repeat the experiment.

R. Washing was done manually following kit package insert instructions. We used the Labsystems Multiskan RC spectrophotometer.

Line 106: The enzyme conjugate was HRP?

R. Yes, it is a HRP. We use a commercial kit, the Clarus Coccidiodes Ab EIA. REF CAB102.

Line 114: How were indeterminate results classified in regard to diagnostic performance?

R. We considered indeterminate as negative. We added this explanation at the material and methods section.

Line 116: Since the purpose of this manuscript is to report a novel LFA, it would be useful for the reader to know if it is an FDA-cleared or approved test. If it’s FDA-registered, but not cleared or approved, that would be useful. Is it CE marked?  Would help the reader determine the potential commercial availability in their country. In other words, has any regulatory body assessed the risk of using this medical device?

R. The product is FDA-cleared for sera and CSF. It is not CE marked. We added this information.

Line 153: How do you define ‘weak’ bands. Was there some objective measure or scoring system to which the reader can refer? Otherwise, it’s unclear how this is useful information.

R. There was not a scoring system. ID were interpreted by a lab professional highly trained. Bands were defined based on professional criteria.

Line 154: PPV and NPV are of little value in this study since samples were hand selected and potentially don’t represent the prevalence of disease in a real-world scenario. Consider calculating PPV and NPV based on anticipated prevalence in a tested population – ideally supported by some data.

R. There is no prior information about disease prevalence.

Line 175: Potentially a better name for the assay would ‘Coccidioides antibody ID’ since the analyte detected are anti-Cocci antibodies.

R. Thank you for the correction.

Line 209: The term “highly accurate” to describe LFAs is vague. Can you support this with a reference(s)?

R. We corrected it.

Line 212: ‘Diagnostic performance’ is a better descriptor than ‘analytical performance’.

R. Agree, we changed it.

Line 215: The fact that the Contreas et al. data was not published in a peer-reviewed journal while the Donovan et al. data was – should be made clear here.

R. We clarified it.

Line 225: As mentioned above the form of cocci should be reported in the results. The Discussion is not the place to provide new information to the reader.

R. We added the clinical form in the new table (Table 1).

Line 247: Diagnostic vs. Analytical performance.

R. we corrected it following your recommendation.

Line 268: The first sentence claims there are no COIs. The second sentence clearly describes a COI.

R. We corrected it.

Author Response

Major comments

This manuscript describes a study to compare the results of Coccidioides LFA with other tests for Ab. The design and analysis is appropriate. The results and conclusions should be limited to the stated objective of the project. It would be nice to include a more thorough comparison of your LFA results seen in other populations, and why there may be differences.

R. Thank you.

Title: Validation is likely not the appropriate term.

R. We changed as “Evaluation of the Analytical Diagnostic Performance”.

Lines 36-37: “Contaminated” is redundant in this sentence.

R. We removed it.

Lines 55 – 58: The last phrase is about the limitations of the Ab tests, but there are two other phrases in between about other types of testing. The way it reads now, it sounds like you’re including culture and microscopy in the Ab tests.

R. We rewrote it.

Lines 62 – 64: This needs to be rewritten. As it is, it states that there are 25+17+8 cases, rather than the 17+8=25.

R. Thank you. We rewrote it.

Lines 63-64: 8 rather than eight.

R. We changed.

Lines 79-82: Needs to be rewritten because it sounds like you have two separate control groups (either clarify here or above).

R. Thank you, we edited it.

Line 109: in the dark (or in darkness) rather than at dark.

R. We corrected it.

Figure 1: The flow chart was helpful.

R. Thank you.

Figure 3: The stated goal of the study is to compare LFA to other tests for Coccidioides. The comparison of eg. EIA vs CIE isn’t relevant. You weren’t asking which is the best, just how does LFA perform in comparison.

R. We would like to present all the data.

Line 205: Most of your cases were proven. Why would your cohort be different and less reliant on Ab tests?

R. We hope the inclusion of this rapid Ab detection assay will impact on cases detection and reduction of turnaround of results. These cases are coming from regions were laboratory infrastructure is limited and the level of biosafety is often not adequate for microbiological diagnosis.

Line 217: Donovan, et al is only reference 22.

R. Thank you, we fixed it.

Lines 224-225: That probably answers the question above, but you should address this.

R. Thank you, we fixed it.

Lines 226-230: Based on the objective of this study, these data should be presented in comparison with the LFA, not on their own.

R. We are sorry, but we did not understand your question.

Line 240: a reason

R. Thank you. We corrected it.

Line 244: “other”

R. Thank you. We corrected it.

Lines 250-252: Is reliance on the “probable” cases also potentially a limitation?

R. We added the limitation. Thank you.

Line 254: My understanding is that validation refers to a specific, rigorous series of tests. You compared the LFA but did not validate it.

R. Yes, based on your comment we changed the validation for the evaluation of the test performance.

Round 2

Reviewer 1 Report

Comments and Suggestions for Authors

The authors have addressed some, but not all, necessary revisions.  See comments below.

Line 65: It is still not clear how the samples that were included were selected from the larger biobank. Potential source of selection bias that needs to discussed. Not adequately addressed  in R1.

Line 65: It is not appropriate to include multiple samples from the same patient for a study describing diagnostic performance.  This was stated in the first review and has not been adequately addressed in R1.

Line 67: The EORTC guidelines require 2-fold rise in antibody titer in serum on consecutive tests to be considered probable coccidioidomycosis. Please provide data supporting this or remove reference to EORTC guidelines for probable cases. The IDSA document appears to be specific for CAP, which only accounts for a subset of the cases included in this study. It also recommends confirmation of EIA with ID or CF.  It should be clear to the reader that probable cases do not strictly adhere to either definition referenced in the manuscript.

Line 91: It doesn’t appear the cases included as non-cocci fungal infections truly fit the definition found in EORTC. The EORTC document does not include serology for Histo or Paracocci. It does include antigen testing for Histo. It’s unclear if the Asper cases included fit the EORTC guidelines.  Please share more data or remove reference to the EORTC guidelines for these cases.  The IDSA guidelines don't include Paracocci or Asper and it’s unclear if  Histo cases fit the IDSA definitions.

Line 140: Please provide 510K number so this can be assessed by journal before publication. I cannot find the LFA submission in the FDA 510k database. I did find that it was determined to be exempt by the manufacturer, so it has been listed (not cleared or approved). If that is the case, this is not a ‘FDA-cleared’ device and has not been assessed by the FDA for safety/efficacy or for essential equivalency to a predicate device. If not truly FDA-cleared please revise manuscript as to not be misleading to the reader.

Line 175: Reference to weak bands should be removed. It suggests the user might be able to differentiate false positives from true positives based on the band intensity but this study is not designed to demonstrated this. It is misleading to the reader. Not adequately addressed in R1.

Line 193: Predictive values should not be reported since disease prevalence is completely artificial/contrived. Not adequately addressed in R1.

Line 273: The unblinded testing of samples, some of which required tests that use manual interpretation open to bias should be discussed as a limitation.

Comments on the Quality of English Language

None

Author Response

Line 65: It is still not clear how the samples that were included were selected from the larger biobank. Potential source of selection bias that needs to discussed. Not adequately addressed  in R1.

R. There was no selection of specimens. We tested all specimens available in the biorepository. We restructured the specimen description in the material and methods section.

About specimens selection, please see the description presented at the material and methods section: “We tested all available sera from patients with proven and positive antibody testing for coccidioidomycosis, and sera from patients with positive antibody testing against other fungi, Histoplasma, Paracoccidioides, and Aspergillus, as specificity controls. Prior to EIA and LFA testing, specimens were retested by ID and CIE methods to validate specimen stability, and the presence of anti-Coccidioides antibodies. (Figure 1).

Line 65: It is not appropriate to include multiple samples from the same patient for a study describing diagnostic performance.  This was stated in the first review and has not been adequately addressed in R1.

R. Thank you. Based on your comment we focus this manuscript on the evaluation of the analytical performance of the test only. We removed the diagnostic component of this analysis. In that way, we decided to use all specimens available aimed to improve the sample size.

Line 67: The EORTC guidelines require 2-fold rise in antibody titer in serum on consecutive tests to be considered probable coccidioidomycosis. Please provide data supporting this or remove reference to EORTC guidelines for probable cases. The IDSA document appears to be specific for CAP, which only accounts for a subset of the cases included in this study. It also recommends confirmation of EIA with ID or CF.  It should be clear to the reader that probable cases do not strictly adhere to either definition referenced in the manuscript.

R. Thank you for your recommendation, we removed the EORTC and IDSA case definitions/references. We restructure the description of specimen’s statement. All specimens were tested simultaneously by ID, CIE and EIA.

Line 91: It doesn’t appear the cases included as non-cocci fungal infections truly fit the definition found in EORTC. The EORTC document does not include serology for Histo or Paracocci. It does include antigen testing for Histo. It’s unclear if the Asper cases included fit the EORTC guidelines.  Please share more data or remove reference to the EORTC guidelines for these cases.  The IDSA guidelines don't include Paracocci or Asper and it’s unclear if  Histo cases fit the IDSA definitions.

R. Thank you, we removed the references.

Line 140: Please provide 510K number so this can be assessed by journal before publication. I cannot find the LFA submission in the FDA 510k database. I did find that it was determined to be exempt by the manufacturer, so it has been listed (not cleared or approved). If that is the case, this is not a ‘FDA-cleared’ device and has not been assessed by the FDA for safety/efficacy or for essential equivalency to a predicate device. If not truly FDA-cleared please revise manuscript as to not be misleading to the reader.

R. We reached IMMY regulatory team, and you are correct, the product is FDA listed. We prefer to remove any information about product classification/registration. This could change, and that is something the reader could investigate.

Line 175: Reference to weak bands should be removed. It suggests the user might be able to differentiate false positives from true positives based on the band intensity but this study is not designed to demonstrated this. It is misleading to the reader. Not adequately addressed in R1.

R. Thank you. We removed the statement.

Line 193: Predictive values should not be reported since disease prevalence is completely artificial/contrived. Not adequately addressed in R1.

R. Agree. We removed the predictive values.

Line 273: The unblinded testing of samples, some of which required tests that use manual interpretation open to bias should be discussed as a limitation.

R. Thank you. We added the limitation listed.

Round 3

Reviewer 1 Report

Comments and Suggestions for Authors

All suggested revisions have been addressed except including multiple samples from a single patient.  We've had a couple rounds of revisions and only including 1 sample / patient been recommended each time. I'll leave it up to the section/editor(s) whether including these samples is appropriate.

Comments on the Quality of English Language

Editing needed